# Biological Activity of *Pogostemon cablin* Essential Oil and Its Potential Use for Food Preservation



Lucia Galovičová [1,*](ID), Petra Borotová [2](ID), Veronika Valková [1](ID), Hana Ďúranová [2](ID), Jana Štefániková [2](ID),
Nenad L. Vukovic [3], Milena Vukic [3](ID) and Miroslava Kačániová [1,4,*](ID)

1 Institute of Horticulture, Faculty of Horticulture and Landscape Engineering,
Slovak University of Agriculture, Tr. A. Hlinku 2, 94976 Nitra, Slovakia; veronika.valkova@uniag.sk
2 AgroBioTech Research Centre, Slovak University of Agriculture, Tr. A. Hlinku 2, 94976 Nitra, Slovakia;
petra.borotova@uniag.sk (P.B.); hana.duranova@uniag.sk (H.Ď.); jana.stefanikova@uniag.sk (J.Š.)
3 Department of Chemistry, Faculty of Science, University of Kragujevac, 34000 Kragujevac, Serbia;
nvchem@yahoo.com (N.L.V.); milena.vukic@pmf.kg.ac.rs (M.V.)
4 Department of Bioenergy, Food Technology and Microbiology, Institute of Food Technology and Nutrition,
University of Rzeszow, 4 Zelwerowicza Str., 35-601 Rzeszow, Poland
* Correspondence: l.galovicova95@gmail.com (L.G.); miroslava.kacaniova@gmail.com (M.K.)

**Abstract:** This study aimed to analyze the biological activity of the essential oil *Pogostemon cablin* (PCEO) to determine the antioxidant, antimicrobial, antibiofilm, insecticidal activity, and chemical composition of the essential oil. We analyzed the structure of biofilms on various surfaces using the MALDI-TOF MS Biotyper and evaluated the antimicrobial effect of the vapor phase of the essential oil in a food model. We determined the main volatile components of PCEO as patchouli alcohol 31.0%, α-bulnesene 21.3%, and α-guaiene 14.3%. The free radical scavenging activity was high and reached $71.4 \pm 0.9\%$, corresponding to $732 \pm 8.1$ TEAC. The antimicrobial activity against bacteria was weak to moderate. We recorded strong activity against yeast. The antifungal activity was very weak in the contact application. Biofilm-producing bacteria were moderately inhibited by PCEO. The change in biofilm structure due to essential oil was demonstrated by MALDI-TOF MS Biotyper analysis. Vapor phase application in a food model showed relatively strong effects against bacteria and significantly higher antifungal efficacy. The insecticidal activity was observed only at higher concentrations of essential oil. Based on the findings, PCEO can be used in the food industry as an antifungal substance in extending the shelf life of bakery products and as protection in the storage of root vegetables.

**Keywords:** *Pogostemon cablin* oil; DPPH; vapor phase; insecticidal activity; antimicrobial activity; antifungal activity; antibiofilm activity; MALDI-TOF MS Biotyper

## 1. Introduction

*Pogostemon cablin* is an aromatic herb native to the Philippines, belonging to the *Lamiaceae* family. This plant is also known as patchouli. *P. cablin* is of great commercial importance. It is widely used in the pharmaceutical and cosmetics industries [1,2]. The use of dried leaves as by-products from these industries in order to preserve sustainable agriculture leads to new techniques of patchouli essential oil production and processing methods [3,4]. *P. cablin* is considered a plant with huge commercial potential due to its unique taste, aromatic properties, and biological activities. *P. cablin* has insecticidal, antibacterial, and antifungal properties [5,6].

Bacteria generate bacterial communities for the same reasons as other organisms; for example, protection from predators or other external hazards, access to nutrients, and genetic diversity [7]. The bacterial community in the form of a biofilm is the aggregation of microbial cells on the surface, which are coated with a matrix of extracellular polymeric substances. The biofilm community increases the resistance of bacteria compared to single

living microorganisms [8]. Biofilms are more resistant to physical influences such as ultraviolet radiation, extreme temperatures, and pH changes. They have a stronger resistance to oxidative stress as well as disinfectants and antibiotics. Due to higher resistance, the persistence of foodborne pathogens may occur in the food industry [9].

*Salmonella enteritidis*, a major foodborne enteric pathogen, forms biofilms on materials of different natures and growth conditions [10]. *Salmonella enteritidis* adheres easily to surfaces and produces biofilm on food contact surfaces and equipment, such as glass, plastic, wood, rubber, or stainless steel, which are commonly used in the food industry [11].

*Pseudomonas fluorescens* is a major food spoilage microorganism that usually occurs in the form of biofilms [12]. *Pseudomonas fluorescens* is used as a model microorganism to study biofilms. It can form biofilm on various biotic and abiotic surfaces, both hydrophobic and hydrophilic such as glass, wood, plastic, and stainless steel [13,14].

In clinical microbiology laboratories, methods based on selection media and biochemical and phenotypic methods are most often used to identify microorganisms. With these methods, the identification of biofilm-producing bacteria is challenging. Rapid and accurate identification of microorganisms is essential in clinical microbiology. For this reason, the simple identification of bacteria and fungi using the MALDI-TOF MS Biotyper has become a revolution [15]. Using the MALDI-TOF MS Biotyper, it is possible to analyse molecular changes in the structure of biofilms over time [16].

Essential oils have been used for centuries in various sectors of the food, pharmaceutical, and cosmetic industries. Currently, essential oils are of scientific and popular interest because they can act synergistically with other preservation techniques, are generally recognized as safe, and have antioxidant, antibacterial, antidiabetic, antimutagenic, insecticidal, non-toxic, and antifungal properties that are promising for their use as bioactive compounds in various foods [17].

This study aimed to analyse the biological activity of PCEO to determine the antioxidant, antimicrobial, antibiofilm activity, insecticidal, and chemical composition of the essential oil. In addition, changes in the biofilm structure on glass and wood surfaces on *Salmonella enteritidis* and *Pseudomonas fluorescens* were evaluated using MALDI-TOF MS Biotyper. We also focused on the antibacterial and antifungal effect of the vapor phase of the essential oil in the food model.

## 2. Materials and Methods

### 2.1. Essential Oil

*Pogostemon cablin* essential oil was purchased from Hanus, s.r.o. (Nitra, Slovakia). The essential oil was obtained by steam distillation of the fermented leaves, followed by maturation of the essential oil over time.

### 2.2. Chemical Composition

PCEO was analyzed by gas chromatography/mass spectrometry (GC/MS) and gas chromatography (GC-FID) GC/MS. Analysis of PCEO was performed using an Agilent 6890N gas chromatograph (Agilent Technologies, Santa Clara, CA, USA) coupled to a 5975B quadrupole mass spectrometer (Agilent Technologies, Santa Clara, CA, USA). HP-5MS capillary column (30 m $\times$ 0.25 mm $\times$ 0.25 $\mu$m) was used. The temperature program was set from 60 °C to 150 °C (increase rate 3 °C/min) and from 150 °C to 280 °C (increase rate 5 °C/min). The total duration of the program was 60 min. Helium 5.0 was used as the carrier gas at a flow rate of 1 mL/min. The injection volume was 1 $\mu$L (sample EO was diluted in pentane), setting the temperature of the split/splitless injector at 280 °C. The sample under investigation was injected in a split mode with a split ratio of 40.8:1. Electron impact mass spectrometry (EI-MS; 70 eV) data were obtained in scan mode in the $m/z$ 35–550 range. The MS sources of the ion source and the MS of the quadrupole were 230 °C and 150 °C, respectively. Data acquisition began after a 3-min solvent delay. GC-FID analyzes were performed on an Agilent 6890N gas chromatograph (Agilent Technologies, Santa Clara, CA, USA) connected

to an FID detector. The column (HP-5MS) and chromatographic conditions were the same as for GC-MS. The FID detector temperature was set at 300 °C.

The individual volatile components of the PCEO sample were identified according to their retention indices [18] and compared with reference spectra (Wiley and NIST databases). Retention indices were determined experimentally by a standard method that included retention times of n-alkanes (C6–C34) injected under the same chromatographic conditions [19]. Percentages of identified compounds (amounts greater than 0.1%) were derived from their GC peak areas.

### 2.3. Determination of Antioxidant Activity

The antioxidant activity of PCEO was determined using 2,2-diphenyl-1-picrylhydrazyl (DPPH, Sigma Aldrich, Schnelldorf, Germany). DPPH stock solution (0.025 g/L dissolved in methanol) was adjusted to an absorbance of 0.8 at 515 nm. Then, 5 μL of PCEO was added to 195 μL of DPPH solution in a 96-well microtiter plate. The reaction mixture was incubated for 30 min in the dark with continuous shaking at 1000 rpm. Antioxidant activity was expressed as a percentage of DPPH inhibition and was calculated according to the formula $(A0 − AA)/A0 × 100$, where $A0$ was the absorbance of DPPH and $AA$ was the absorbance of the sample.

Radical scavenging activity was recalculated against a standard reference substance Trolox (Sigma Aldrich, Schnelldorf, Germany) dissolved in methanol (Uvasol® for spectroscopy, Merck, Darmstadt, Germany) to a concentration range of 0–100 μg/mL. The total radical scavenging capacity was expressed according to the calibration curve as 1 μg Trolox per 1 mL essential oil sample (TEAC).

### 2.4. Microorganisms

Gram-negative bacteria (*Pseudomonas aeruginosa* CCM 3955, *Yersinia enterocolitica* CCM 7204, *Salmonella enterica* subsp. *enterica* ser. Enteritidis CCM 4420, *Serratia marcescens* CCM 8587), gram-positive bacteria (*Bacillus subtilis* CCM 1999, *Staphylococcus aureus* subsp. *aureus* CCM 8223, *Enterococcus faecalis* CCM 4224, *Micrococcus luteus* CCM 732) and yeasts (*Candida krusei* CCM 8271, *Candida albicans* CCM 8261, *Candida tropicalis* CCM 8223, *Candida glabrata* CCM 8270) were obtained from the Czech Collection of Microorganisms (Brno, Czech Republic). The biofilm-forming bacterial strain *Pseudomonas fluorescens* was isolated from fish and *Salmonella enteritidis* was isolated from a meat sample. The biofilm bacterial strains were identified by 16S rRNA sequencing and MALDI-TOF MS Biotyper. *Penicillium aurantiogriseum*, *Penicillium chrysogenum*, *Penicillium expansum* and *Penicillium italicum* were obtained from grape samples and identified by 16S rRNA sequencing and MALDI-TOF MS biotype.

### 2.5. Determination of Antimicrobial Activity

The antimicrobial activity of PCEO was determined by the disk diffusion method. The inoculum was cultured for 24 h on Tryptone Soy Agar (TSA, Oxoid, Basingstoke, UK) at 37 °C for bacteria and on Sabouraud Dextrose Agar (SDA, Oxoid, Basingstoke, UK) at 25 °C for yeast. The inoculum was adjusted to an optical density of 0.5 McFarland standard $(1.5 × 10^8$ CFU/mL). In addition, 100 μL of conditioned inoculum was applied to a Petri dish (PD) with Mueller Hinton agar (MHA, Oxoid, Basingstoke, UK) for bacteria and with SDA for yeasts. Sterile 6 mm disks were placed on PD with tweezers. Then, 10 μL of PCEO was applied to the disks. The samples were incubated for 24 h at 37 °C for bacteria and 25 °C for yeast. Antibiotics (cefoxitin, gentamicin, Oxoid, Basingstoke, UK) were used as a positive control for gram-negative and gram-positive bacteria. An antifungal (fluconazole, Oxoid, Basingstoke, UK) was used as a positive control for yeast. Disks impregnated with 0.1% DMSO (dimethyl sulfoxide, Centralchem, Bratislava, Slovakia) served as a negative control.

An inhibition zone above 10 mm was determined to be very strong antimicrobial activity, an inhibition zone above 5 mm was determined to be mild activity, and an inhibition

zone above 1 mm was determined to be weak activity. Antimicrobial activity was measured three times.

### 2.6. Minimum Inhibitory Concentration (MIC)

The MICs of bacteria and yeasts were determined using the agar microdilution method. The inoculum was cultured for 24 h in Mueller Hinton Broth (MHB, Oxoid, Basingstoke, UK) at 37 °C for bacteria and Sabouraud Dextrose Broth (SDB, Oxoid, Basingstoke, UK) at 25 °C for yeast. Then, 100 µL of nutrient medium and 50 µL of inoculum with an optical density of 0.5 McFarland standard were applied to a 96-well microtiter plate. Subsequently, PCEO was prepared by serial dilution to a concentration range of 400 µL/mL to 0.2 µL/mL in MHB/SDB and mixed thoroughly with bacterial inoculum in the wells. The prepared 96-well microtiter plates were measured at 570 nm with a Glomax spectrophotometer (Promega Inc., Madison, WI, USA) at 0 h. Subsequently, the bacterial samples were incubated at 37 °C for 24 h. Yeast samples were incubated at 25 °C for 24 h and measured again. MHB/SDB with essential oil was used as a negative control, and MHB/SDB with inoculum was used as a positive control for maximal growth.

The absorbance of the biofilm-forming bacteria was measured using crystal violet. In the first step, the unbound cell suspension was discarded, and the wells were washed three times with distilled water and allowed to dry at room temperature. Subsequently, 200 µL of 0.1% (*w/v*) crystal violet was added to the wells and the samples were incubated for 15 min at room temperature. After the incubation, the wells were washed repeatedly and dried. Stained biofilms were resolubilized with 200 µL of 33% acetic acid [20]. The absorbance was measured at 570 nm.

The concentration of essential oil where the change in absorbance after 24 h was lower than the Δ in absorbance of the control sample (maximal growth of microorganism) was determined as the minimum inhibitory concentration. The test was performed in triplicate. Minimum inhibitory concentrations of microscopic fibrous fungi of the genus *Penicillium* were determined by the disk diffusion method. The inoculum was cultured for 7 days at 25 °C on SDA agar. The inoculum was adjusted to an optical density of 0.5 McFarland standard. PCEO concentrations (500, 250, 125, 62.5 µL/L) were prepared by dilution in 0.1% DMSO. Then, 100 µL of inoculum was plated on PD with SDA agar. Subsequently, 6 mm sterile paper disks were placed on agar with tweezers, and 10 µL of the appropriate concentration of PCEO was applied to the disks. The samples were incubated for 7 days at 25 °C. An inhibition zone above 15 mm was determined very strong antifungal activity, an inhibition zone above 10 mm was determined mild activity, and an inhibition zone above 5 mm was determined weak activity. The analysis was performed in triplicate.

### 2.7. Analysis of Differences in Biofilm Development with MALDI-TOF MS Biotyper

Changes in the molecular structure of protein spectra during biofilm development after the addition of PCEO were evaluated by MALDI-TOF MS Biotyper (Bruker, Bremen, Germany). The analysis was performed in 50 mL polypropylene centrifuge tubes. Approximately 20 mL of MHB, a wooden toothpick, and a glass slide were added to the tubes. The inoculum biofilm-producing bacteria were cultured at 37 °C for 24 h. The inoculum was adjusted to an optical density of 0.5 McFarland standard and 100 µL was added to the tubes. PCEO was added to the experimental groups at a concentration of 0.1% *v/v*, and the control groups remained untreated. The samples were incubated at 37 °C on a shaker at 170 rpm.

Using a sterile cotton swab, biofilm was removed from both test surfaces and pressed onto a MALDI-TOF MS Biotyper metal target plate (Bruker, Bremen, Germany). Planktonic cells were isolated from the culture medium. Subsequently, 300 µL of culture medium was centrifuged for 1 min at 12,000 rpm. The supernatant was discarded, and the pellet was washed repeatedly in 30 µL of ultrapure water. In the last step, the pellet was resuspended, and 1 µL was applied to a MALDI-TOF MS Biotyper metal target plate (Bruker, Bremen, Germany).

The target plate was dried, and 1 μL of the α-cyano-4-hydroxycinnamic acid matrix (10 mg/mL) was applied to the surface of the samples. Samples were processed by MALDI-TOF MicroFlex (Bruker Daltonics, Bremen, Germany) linear and were positive mode for the range *m/z* 200–2000 after crystallization. The spectra were obtained by automated analysis and the same sample similarities were used to generate the standard global spectrum (MSP). In turn, 19 MSPs were generated from the spectra using MALDI Biotyper 3.0 (Bruker, Bremen, Germany) and were grouped into dendrograms using Euclidean distance [21]. Samples were analyzed after 3, 5, 7, 9, 12, and 14 days.

### 2.8. In Situ Antimicrobial Analysis on a Food Model

The antimicrobial effect of the PCEO vapor phase was evaluated in 0.5 L sterile glass jars (Bormioli Rocco, Parma, Italy) on bread used as a food model. *Penicillium* fungi were cultured for 7 days on SDA at 25 °C. *S. marcescens* and *M. luteus* were cultured for 24 h on TSA at 37 °C. Cultures were applied to bread slices (15 × 15 × 1.5 cm) with three pokes. A 6 cm sterile filter paper was placed on the lid of the container. Then, 100 μL of PCEO (62.5, 125, 250, and 500 μL/L diluted in ethyl acetate) were applied to the filter paper. The control group was left untreated. The dishes were hermetically sealed and incubated in the dark for 14 days at 25 °C ± 1 °C for fibrous microscopes and 7 days at 37 °C for bacteria.

In situ antimicrobial analysis in the vapor phase on a vegetable model (carrot, celery) was tested for *S. marcescens*. Warm SDAs for microscopic fibrous fungi and MHAs for bacteria were poured into 60 mm PD and PD's lids. Chopped vegetables (0.5 mm) were placed on agar. The inoculum was then prepared as described in the above section. PCEO was diluted in ethyl acetate to concentrations of 500, 250, 125, and 62.5 μL/L. The sterile filter paper was placed into the PD lid, and 100 μL of the appropriate concentration of essential oil was applied. The lid was left open for 1 min to evaporate the remaining ethyl acetate, then the plates were sealed and incubated at 37 °C for 7 days with bacteria and for 14 days at 25 °C with fibrous microscopic fungi.

Fungal/mycelial growth inhibition was evaluated by stereological methods. The bulk density (Vv) of the fungi was estimated using ImageJ software. The stereological lattice of the colonies (P) and the substrate (p) were calculated. Fungal growth density was calculated in % according to the formula Vv = P/p × 100. Antifungal activity EO was expressed as inhibition of mycelial growth in % (MGI): MGI = [(C − T)/C] × 100, where C was the density fungal growth in the control group and T was the fungal growth density in the treated group [22].

In situ bacterial growth was determined using stereological methods. In this concept, the bulk density (Vv) of the bacterial colonies was first estimated using ImageJ software and counting the stereological lattice points affecting the colonies (P) and those (p) falling into the reference space (growth substrate used). The bulk density of the bacterial colonies was then calculated as follows: Vv (%) = P/p. EO antibacterial activity was defined as the percentage of bacterial growth inhibition (BGI) BGI = [(C − T)/C] × 100, where C and T were bacterial growth (expressed as Vv) in the control and treatment groups, respectively. Negative results represented a growth stimulation of microorganisms.

### 2.9. Insecticidal Activity

The insecticidal activity of PCEO was evaluated on a model organism, *Pyrrhocoris apterus*. Fifty *P. apterus* individuals were placed in the PD. A circle of sterile filter paper was glued to the lid. Concentrations (50, 25, 12.5, 6.25, and 3.125%) were prepared by diluting PCEO with 0.1% polysorbate. Subsequently, 100 μL of the appropriate concentration of PCEO was applied to the sterile filter paper. The dishes were sealed around the perimeter with parafilm and left at room temperature for 24 h. In the control group, 100 μL of 0.1% polysorbate was used. After 24 h, the number of living and dead individuals was evaluated. The experiment was performed in triplicate.

### 2.10. Statistical Data Processing

One-way analysis of variance (ANOVA) was performed using Prism 8.0.1 (GraphPad Software, San Diego, CA, USA) followed by Tukey's test at $p < 0.05$. SAS® software version 8 was used for data processing. The results of the MIC value (concentration that caused 50% and 90% inhibition of bacterial growth) were determined by logit analysis.

## 3. Results

### 3.1. Chemical Composition

Based on chromatographic/mass spectrometry (GC/MS) and gas chromatography (GC-FID) analysis, we determined that PCEO contained 31.0% patchouli alcohol, 21.3% α-bulnesene, 14.3% α-guaiene, and 6.9% seychellene as the main volatile components. (Table 1).

**Table 1.** Chemical composition of PCEO.

| RI [a] | Compound [b] | % [c] |
|---|---|---|
| 938 | a-pinene | tr |
| 980 | b-pinene | 0.1 ± 0.01 |
| 1028 | a-limonene | tr |
| 1383 | b-patchoulene | 3.0 ± 0.03 |
| 1388 | b-elemene | 2.2 ± 0.02 |
| 1408 | a-gurjunene | 0.6 ± 0.03 |
| 1422 | (*E*)-caryophyllene | 3.4 ± 0.10 |
| 1440 | a-guaiene | 14.3 ± 0.4 |
| 1445 | Seychellene | 6.9 ± 0.10 |
| 1456 | a-humulene | 0.7 ± 0.03 |
| 1458 | a-patchoulene | 6.8 ± 0.10 |
| 1490 | b-selinene | 2.2 ± 0.05 |
| 1498 | Ledene | 0.4 ± 0.02 |
| 1499 | a-selinene | 3.3 ± 0.02 |
| 1512 | a-bulnesene | 21.3 ± 0.63 |
| 1583 | caryophyllene oxide | 1.1 ± 0.04 |
| 1596 | Globulol | 0.9 ± 0.02 |
| 1609 | 5-epi-7-epi-a-eudesmol | 1.1 ± 0.01 |
| 1663 | patchouli alcohol | 31.0 ± 1.46 |
| | Total | 99.2 ± 2.07 |

[a] Values of retention indices on on HP-5MS column; [b] Identified compounds; [c] tr—compounds identified in amounts less than 0.1%.

### 3.2. Antioxidant and Antimicrobial Activity

The antioxidant activity of PCEO was determined using the DPPH radical. The free radical scavenging activity reached 71.4 ± 0.9%, which corresponds to 732 ± 8.1 µg of TEAC/mL of sample. Using the disk diffusion method, we determined the zones of inhibition and subsequently the inhibitory activity of the essential oil (Table 2). In gram-negative bacteria, we observed weak antimicrobial activity against *P. aeruginosa* and *S. enterica*, and moderate activity against *Y. enterocolitica* and *S. marcescens*. In gram-positive microorganisms, we detected weak antimicrobial activity against *B. subtilis* and *S. aureus*, and moderate antimicrobial activity was shown against *E. faecalis* and *M. luteus*. We observed strong antimicrobial activity of PCEO against all tested yeast species. We observed moderately weak antimicrobial activity against biofilm-producing bacteria.

### 3.3. Minimum Inhibitory Concentration

Using the agar microdilution method, we determined the minimum inhibitory concentrations for bacteria and yeasts (Table 3). We recorded the lowest MIC values against the yeast and biofilm-producing bacteria, *S. enteritidis*. Moderate/medium values were detected against gram-positive microorganisms. Higher MIC values were observed for gram-negative microorganisms. We analyzed the minimum inhibitory concentrations of

fibrous microscopic fungi using the disk diffusion method. For microscopic filamentous fungi, we observed the lowest minimum inhibitory concentration (Table 4) in *P. aurantiogriseum* (62.5 µL/L). For *P. chrysogenum* and *P. expansum*, we recorded a MIC of 125 µL/L. The highest MIC of 250 µL/L was detected in *P. italicum*.

**Table 2.** Antimicrobial activity of PCEO.

| Microorganism | Zone Inhibition (mm) | Activity of EO | ATB |
|---|---|---|---|
| *Pseudomonas aeruginosa* | 1.00 ± 0.00 | * | 22 ± 1.00 |
| *Yersinia enterocolitica* | 6.67 ± 0.58 | ** | 25 ± 2.00 |
| *Salmonella enterica* | 1.67 ± 0.58 | * | 25 ± 1.50 |
| *Serratia marcescens* | 6.67 ± 1.15 | ** | 27 ± 2.00 |
| *Bacillus subtilis* | 2.00 ± 0.00 | * | 31 ± 3.00 |
| *Staphylococcus aureus* | 4.33 ± 0.58 | * | 31 ± 1.00 |
| *Enterococcus faecalis* | 5.67 ± 0.58 | ** | 28 ± 0.50 |
| *Micrococcus luteus* | 6.67 ± 1.15 | ** | 26 ± 2.00 |
| *Candida krusei* | 19.00 ± 1.00 | *** | 31 ± 3.00 |
| *Candida albicans* | 16.00 ± 1.00 | *** | 25 ± 2.00 |
| *Candida tropicalis* | 17.67 ± 0.58 | *** | 31 ± 1.00 |
| *Candida glabrata* | 13.00 ± 1.73 | *** | 31 ± 1.50 |
| *Pseudomonas fluorescens* biofilm | 5.33 ± 1.53 | * | 26 ± 1.00 |
| *Salmonella enteritidis* biofilm | 5.33 ± 1.15 | ** | 25 ± 1.00 |

* Weak antimicrobial activity (zone 1–5 mm). ** Moderate inhibitory activity (zone 5–10 mm). *** Very strong inhibitory activity (zone > 10 mm), ATB—antibiotics, positive control (cefoxitin for G−, gentamicin for G+, fluconazole for yeast).

**Table 3.** Minimal inhibition concentration of PCEO against for bacteria and yeast.

| Microorganism | MIC 50 (µL/mL) | MIC 90 (µL/mL) |
|---|---|---|
| *Pseudomonas aeruginosa* | 24.53 | 35.18 |
| *Yersinia enterocolitica* | 22.51 | 33.12 |
| *Salmonella enterica* | 22.34 | 25.19 |
| *Serratia marcescens* | 25.64 | 28.93 |
| *Bacillus subtilis* | 11.15 | 12.56 |
| *Staphylococcus aureus* | 8.99 | 10.13 |
| *Enterococcus faecalis* | 12.58 | 15.32 |
| *Micrococcus luteus* | 16.28 | 18.33 |
| *Candida krusei* | 10.15 | 13.43 |
| *Candida albicans* | 8.12 | 9.65 |
| *Candida tropicalis* | 6.98 | 8.93 |
| *Candida glabrata* | 9.12 | 12.38 |
| *Pseudomonas fluorescens* biofilm | 28.53 | 36.58 |
| *Salmonella enteritidis* biofilm | 5.35 | 6.87 |

**Table 4.** Minimal inhibition concentration for fungi.

| | Patchouli EO (µL/L) | | | |
|---|---|---|---|---|
| Fungi Strains | 62.5 | 125 | 250 | 500 |
| *P. chrysogenum* | 0.00 ± 0.00 [a] | 0.50 ± 0.00 [b] | 1.33 ± 0.58 [c] | 4.33 ± 0.58 [d] |
| *P. aurantiogriseum* | 0.80 ± 0.26 [a] | 1.23 ± 0.25 [a] | 1.60 ± 0.53 [a] | 5.43 ± 0.51 [b] |
| *P. expansum* | 0.00 ± 0.00 [a] | 1.17 ± 0.29 [b] | 1.50 ± 0.50 [b] | 3.80 ± 0.72 [c] |
| *P. italicum* | 0.00 ± 0.00 [a] | 0.00 ± 0.00 [a] | 2.50 ± 0.50 [b] | 3.53 ± 0.50 [b] |

Means ± standard deviation. Values followed by different superscript within the same row are significantly different ($p < 0.05$).

### 3.4. Analysis of Biofilm Developmental Phases and Evaluation of Molecular Differences on Different Surfaces Using MALDI-TOF MS Biotyper

The effect of PCEO on the molecular structure and growth inhibition of *P. fluorescens* and *S. enteritidis* biofilms was evaluated using the MALDI TOF MS Biotyper. The spectra of

biofilms and planktonic cells in the control group developed identically, and therefore, the spectra of planktonic cells were used as a control spectrum for greater clarity. For each day, two experimental spectra from different surfaces (glass, wood) and a planktonic spectrum representing the development of the control group are shown.

The mass spectra obtained on the third and fifth days of the experiment represent very small differences in the peaks between the experimental and control plankton spectra (Figure A1 in Appendix A). On days seven and nine of the experiment, we recorded the difference between the experimental wood spectrum and the control spectrum. During days 12 and 14, we observed differences in both experimental groups compared to the control. Under the influence of PCEO, we were able to observe changes in the protein spectrum of the biofilm. This finding suggests that the essential oil disrupts biofilm homeostasis, leading to degradation of this form of the microorganism.

To visualize the similarity of the mass spectra, a dendrogram based on MSP distances was constructed (Figure 1). Control groups and young biofilm spectra had the shortest distance with plankton cells (days 3 and 5). From day seven, we could see an increase in the distance of the experimental groups in the distance of MSPs with a greater distance in the experimental group made of wood. During the 12th and 14th days, it is possible to see more significant MSPs than in the previous days. The distance of MSP control groups from all tested days was shorter than in the experimental groups. The increasing distance of MSP in the experimental groups indicates changes in the protein profile of the bacterial biofilm, *P. fluorescens*.

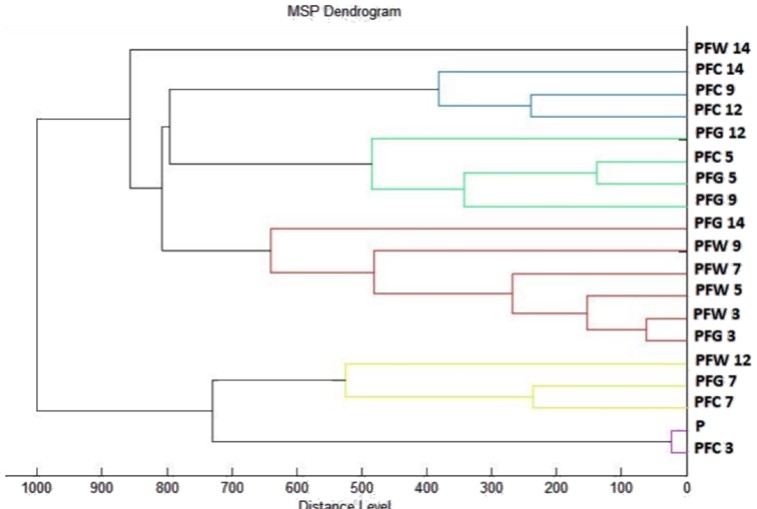

**Figure 1.** Dendrogram of *P. fluorescens* generated using MSPs of the planktonic cells and the control. PF, *P. fluorescens*; C, control; G, glass; W, wood; and P, planktonic cells.

The mass spectra obtained from *S. enteritidis* on the third day of the experiment represent very small differences in peaks between the experimental and control plankton spectra (Figure A2 in Appendix A). From day five, we observed a difference in spectra between the two experimental groups and the control planktonic group. Under the influence of PCEO, we were able to observe changes in the protein spectrum of the *S. enteritidis* biofilm. This finding suggests that the essential oil disrupts biofilm homeostasis, leading to the degradation of this form of the microorganism.

To visualize the similarity of the mass spectra, a dendrogram based on MSP distances was constructed (Figure 2). The control groups and spectra of the young biofilm on the third day had the shortest distance together with the planktonic cells. From day five, we could see an increase in the distance of experimental groups in the distance of MSPs. During the 14th day, it is possible to see a more significant MSP distance than on previous days. The distance of MSP control groups from all tested days was shorter than in the experimental

groups. The increasing distance of MSP in the experimental groups indicates changes in the protein profile of the bacterial biofilm of *S. enteritidis*.

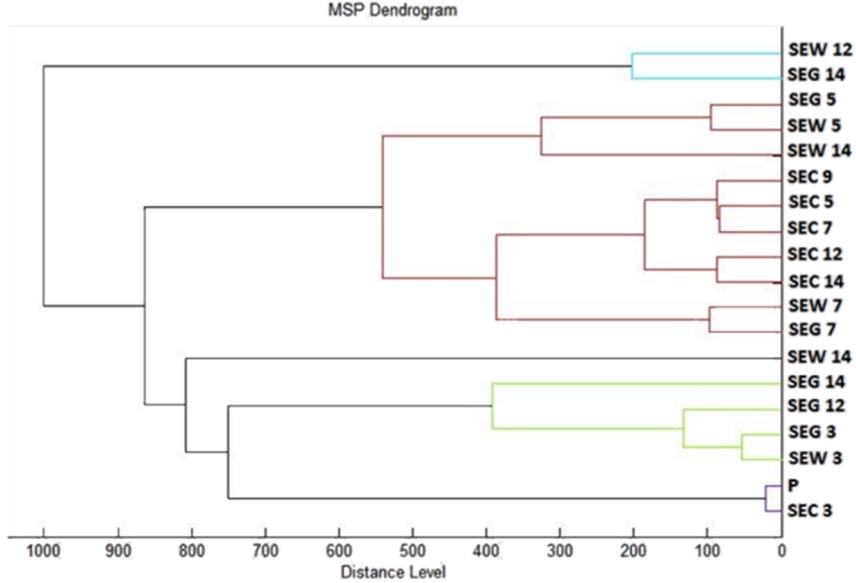

**Figure 2.** Dendrogram of *S. enteritidis* generated using MSPs of the planktonic cells and the control. SE, *S. enteritidis*; C, control; G, glass; W, wood; and P, planktonic cells.

### 3.5. In Situ Antimicrobial Activity in Food Models

The results from in situ evaluation revealed a strong antibacterial activity of patchouli EO in all concentrations applied against the growth of *M. luteus* on bread as a food model. The highest concentration of patchouli EO had also strong inhibitory action on the growth of *S. marcescens* on bread. Against the bacterium, moderate antibacterial effectiveness was exhibited by the lowest concentrations (62.5, and 125 µL/L) of the patchouli EO. Interestingly, the dose of 250 µL/L caused an almost 2-fold increase in the *S. marcescens* growth indicating a promotion in bacterial activity (Table 5).

**Table 5.** In situ analysis of the antibacterial activity of the vapor phase of PCEO in bread.

| | Bacterial Growth Inhibition [%] Bread | | | |
|---|---|---|---|---|
| **Bacteria** | Patchouli EO (µL/L) | | | |
| | **62.5** | **125** | **250** | **500** |
| *Micrococcus luteus* | 74.59 ± 5.56 [a] | 79.10 ± 8.85 [a] | 75.72 ± 6.18 [a] | 79.07 ± 4.25 [a] |
| *Serratia marcescens* | 44.83 ± 6.99 [a] | 49.16 ± 6.39 [a] | −193.82 ± 4.95 [b] | 80.42 ± 5.95[c] |

Mean ± standard deviation. Values followed by different superscript within the same row are statistically different ($p < 0.05$); the negative values indicate a probacterial activity of the essential oil against the growth of bacteria strains.

The antimicrobial activity was also evaluated on a carrot, used as a growth substrate. Only the concentrations 125 and 62.5 µL/L of patchouli EO significantly inhibited the growth of *M. luteus* and *S. marcescens*, respectively (Table 6). On the other hand, the concentrations of 125 µL/L and higher had very strong activity on bacterial growth of *S. marcescens*.

The growth of *M. luteus* and *S. marcescens* on celery was strongly inhibited by patchouli EO in the concentrations of 62.5 and 500 µL/L, respectively. However, the concentrations below 250 µL/L exhibited an increase in the growth of *S. marcescens*, which suggests a growth-promotion effect of patchouli EO (Table 7).

**Table 6.** In situ analysis of the antibacterial activity of the vapor phase of PCEO in carrot.

| Bacteria | Bacterial Growth Inhibition [%] Carrot | | | |
| | Patchouli EO (µL/L) | | | |
| | 62.5 | 125 | 250 | 500 |
|---|---|---|---|---|
| *Micrococcus luteus* | 0.00 ± 0.00 [a] | 67.50 ± 5.53 [b] | 0.00 ± 0.00 [a] | 0.00 ± 0.00 [a] |
| *Serratia marcescens* | 56.85 ± 6.77 [a] | −1533.33 ± 9.55 [b] | −626.92 ± 5.25 [c] | −1045.56 ± 9.65 [d] |

Mean ± standard deviation. Values followed by different superscript within the same row are statistically different ($p < 0.05$); the negative values indicate a probacterial activity of the essential oil against the growth of bacteria strains.

**Table 7.** In situ analysis of the antibacterial activity of the vapor phase of PCEO in celery.

| Bacteria | Bacterial Growth Inhibition [%] Celery | | | |
| | Patchouli EO (µL/L) | | | |
| | 62.5 | 125 | 250 | 500 |
|---|---|---|---|---|
| *Micrococcus luteus* | 53.56 ± 8.64 [a] | 29.18 ± 5.83 [b] | 0.00 ± 0.00 [c] | 0.00 ± 0.00 [c] |
| *Serratia marcescens* | −62.19 ± 6.49 [a] | −48.36 ± 8.69 [a] | −102.31 ± 9.55 [b] | 59.83 ± 4.64 [c] |

Mean ± standard deviation. Values followed by different superscript within the same row are statistically different ($p < 0.05$); the negative values indicate a probacterial activity of the essential oil against the growth of bacteria strains.

Our results revealed the strong antifungal activity of patchouli EO in the concentrations of 125 and 500 µL/L against *P. aurantiogriseum* growth on a bread model (Table 8). The growth of *P. expansum* was moderately inhibited by all the concentrations used. Against the growth of *P. chrysogenum*, only the weak inhibitory action of the concentrations of ≤250 µL/L of the EO was observed. We have also found that the antifungal activity of the EO against the growth of *P. italicum* was increased with increasing its concentration, and the effect was similar between the concentrations of 250 and 500 µL/L.

**Table 8.** In situ analysis of the antifungal activity of the vapor phase of PCEO in bread.

| Fungi | Mycelial Growth Inhibition [%] Bread | | | |
| | Patchouli EO (µL/L) | | | |
| | 62.5 | 125 | 250 | 500 |
|---|---|---|---|---|
| *P. aurantiogriseum* | −32.61 ± 7.01 [a] | 63.53 ± 6.33 [b] | 6.04 ± 4.82 [c] | 57.26 ± 3.77 [b] |
| *P. expansum* | 45.97 ± 5.11 [a] | 41.64 ± 6.44 [a] | 38.48 ± 6.57 [a] | 49.76 ± 5.69 [a] |
| *P. chrysogenum* | 15.48 ± 4.57 [ab] | 22.46 ± 4.82 [a] | 9.35 ± 3.93 [b] | −1.80 ± 4.17 [c] |
| *P. italicum* | 10.90 ± 4.67 [a] | 28.35 ± 5.28 [b] | 48.55 ± 5.39 [c] | 54.47 ± 6.71 [c] |

Mean ± standard deviation. Values followed by different superscript within the same row are significantly different ($p < 0.05$); the negative values indicate a profungal activity of the essential oil against the growth of fungi strains.

The highest concentrations of patchouli EO had very strong antifungal effects against the growth of *P. aurantiogriseum* (250 and 500 µL/L) and *P. italicum* (500 µL/L) which were growing on a carrot (Table 9). The lower concentrations of the EO also exhibited inhibitory actions against the fungal growth (125 µL/L in *P. aurantiogriseum*, and 125 and 250 µL/L in *P. italicum*) but they were significantly weaker. Against the growth of *P. expansum* and *P. chrysogenum* on the carrot, only weak or even no antifungal effectiveness of the EO was reported.

In situ evaluation on a celery model showed the very strong antifungal activity of patchouli EO against the growth of *P. expansum* and *P. chrysogenum* (in all concentrations), and *P. aurantiogriseum* (250 µL/L). Against *P. italicum*, only weak (125 µL/L) to moderate (remaining concentrations used) antifungal activity was revealed (Table 10).

**Table 9.** In situ analysis of the antifungal activity of the vapor phase of PCEO in carrot.

| Fungi | Mycelial Growth Inhibition [%] Carrot | | | |
| | Patchouli EO (µL/L) | | | |
| | 62.5 | 125 | 250 | 500 |
| --- | --- | --- | --- | --- |
| *P. aurantiogriseum* | 48.32 ± 5.73 [a] | 64.95 ± 4.57 [b] | 97.70 ± 7.12 [c] | 100.00 ± 0.00 [c] |
| *P. expansum* | 2.41 ± 4.17 [a] | 4.88 ± 3.97 [a] | 10.68 ± 6.48 [a] | 25.33 ± 4.59 [b] |
| *P. chrysogenum* | 0.00 ± 0.00 [a] | 0.00 ± 0.00 [a] | 0.00 ± 0.00 [a] | 2.56 ± 3.11 [a] |
| *P. italicum* | 27.06 ± 5.25 [a] | 66.37 ± 5.17 [b] | 75.31 ± 6.63 [b] | 91.46 ± 5.88 [c] |

Mean ± standard deviation. Values followed by different superscript within the same row are significantly different ($p < 0.05$); the negative values indicate a profungal activity of the essential oil against the growth of fungi strains.

**Table 10.** In situ analysis of the antifungal activity of the vapor phase of PCEO in celery.

| Fungi | Mycelial Growth Inhibition [%] Celery | | | |
| | Patchouli EO (µL/L) | | | |
| | 62.5 | 125 | 250 | 500 |
| --- | --- | --- | --- | --- |
| *P. aurantiogriseum* | −13.95 ± 4.11 [a] | 34.84 ± 5.55 [b] | 85.24 ± 7.32 [c] | 58.71 ± 4.59 [d] |
| *P. expansum* | 95.51 ± 5.07 [a] | 98.09 ± 4.62 [a] | 97.17 ± 4.15 [a] | 98.48 ± 4.34 [a] |
| *P. chrysogenum* | 95.31 ± 6.51 [a] | 97.85 ± 8.75 [a] | 95.16 ± 4.16 [a] | 98.31 ± 6.41 [a] |
| *P. italicum* | 40.98 ± 7.74 [a] | 21.74 ± 2.42 [b] | 43.98 ± 5.51 [a] | 41.94 ± 4.58 [a] |

Mean ± standard deviation. Values followed by different superscript within the same row are significantly different ($p < 0.05$); the negative values indicate a profungal activity of the essential oil against the growth of fungi strains.

*3.6. Insecticidal Activity*

We evaluated the insecticidal activity of PCEO as weak (Table 11). The insecticidal activity started at EO concentration 12.5%, with insecticidal activity 2.5%, and reached only 7.5% insecticidal activity at the highest EO concentration (50%). Concentrations of essential oil 6.25 and lower did not show any insecticidal effect.

**Table 11.** Insecticidal activity of PCEO.

| Concentration EO | Living Individuals | Dead Individuals | Insecticidal Activity |
| --- | --- | --- | --- |
| Control | 50 | 0 | 0.0% |
| 3.125% | 50 | 0 | 0.0% |
| 6.25% | 50 | 0 | 0.0% |
| 12.5% | 45 | 5 | 2.5% |
| 25% | 40 | 10 | 5.0% |
| 50% | 35 | 15 | 7.5% |

**4. Discussion**

Hu et al. [23] identified *β*-patchoulene, caryophyllene, *α*-guaiene seychelene, *β*-guaiene, 5-guaiene, spathulenol, patchouli alcohol, and pogoston as the main components of PCEO. The authors analyzed PCEOs from various provinces in China. PCEO analyzed by the Slovak company, Hanus s.r.o. (Nitra, Slovakia), was made from plant material collected in Indonesia which may be the reason for the difference in the main chemical components. Kusuma et al. [4], in their study of the GC-MS analysis of PCEO, identified alcohol with 53.68%, *α*-guaiene 11.26%, and azulene 10.75% as the main components. The authors used microwave extraction, which could affect the composition of PCEO in comparison with the distillation method that was used to produce the essential oil we analyzed. Feng et al. [24] determined that the major component of the essential oil was patchouli (51.1%), followed by fluoroacetophenone (23.5%), and *β*-patchoulene (7.3%). The authors' result corresponds to our finding that patchouli alcohol is the dominant component of PCEO along with

β-patchoulen. Floroacetophenone could not be detected in our sample due to differences in climatic and geographical growing conditions. Santos et al. [25] identified GC-MS (6.12%), α-bulnesene (4.11%), norpatchoulenol (5.72%), pogostol (6.33%), and patchouli alcohol (3.25%) by GC-MS analysis. The authors confirm that, as in our study, patchouli alcohol was the dominant component of the essential oil. Paulus et al. [26] identified *P. cablin* paculol (31.5%), seichelene (13.6%), and α-bulnezene (15.6%) as the main components of the essential oil. The authors' results correlate with our findings even though they detected a higher proportion of seichelene than in our sample. In Liu et al. [27], 23 components were determined in PCEO by GC-MS and the main components were patchouli (41.31%), pogoston (18.06%), α-bulnesene (6.56%), caryophyllene (5.96%), and seychelene (4.32%). The authors detected a higher proportion of alcohol patchouli than in our work and, in addition, detected the presence of pogostone, but in the other components, our findings agree. Tsai et al. [28] identified 41 components of PCEO, with α-guaiene (20.62%) and α-bulnezene (16.18%) having the highest proportion. Donelian et al. [29] identified patchouli, δ-guaiene, α-guaiene, α-patchoulene, and β-caryophyllene as the main components of PCEO. The authors found a similar composition as in our work.

Santos et al. [25] found in their work antioxidant activity using the DPPH method more than 50%, and the IC 50 value was 329.81 μg/mL based on the obtained values considering the antioxidant activity of PCEO to be high. Soh et al. [30] determined the IC 50 at various extractions from 0.42 to 1.92 mg/mL and evaluated this activity as excellent. Paulus et al. [26] determined the antioxidant activity of *P. cablin* at 12.08 μmol Trolox/mL. Mansuri et al. [31] detected PCEO free radical scavenging activity with an IC 50 of 19.53 μg/mL. Despite the inconsistencies in the methods used and the expression of antioxidant activity, the authors agree that PCEO has relatively high antioxidant activity. Chakrapani et al. [32] analysed the effect of *P. cablin* by disk diffusion method and determined inhibition zones ranging from 16 to 20.5 mm against *E. coli*, 12 to 19.3 mm against *B. subtilis*, 18 to 21 mm against *S. aureus*, and 10 to 15.8 mm against *E. aerogenes*. Higher zones of inhibition compared to our results were probably obtained due to the application of higher amounts of PCEO to the disk. Pratama et al. [33] tested the antimicrobial activity against *E. coli* and *S. aureus* by disk diffusion and measured inhibition zones of 20.9 mm and 19 mm. The authors used a higher concentration of extract compared to our study, which led to larger inhibition zones. Aisyah et al. [34] established inhibition zones for *S. aureus* (11.36 ± 1.85 mm). The yeast *C. albicans* seemed to be resistant against patchouli EO with an inhibition zone of 0 mm. In our work, in contrast to the authors, we recorded a significantly higher activity of PCEO against yeast and lower for *S. aureus*; these differences may have arisen due to the different chemical profiles of essential oil. Dechayont et al. [35] detected inhibition zones for MRSA (11.67 ± 1.53 mm), *S. aureus* (10.33 ± 2.52 mm), and *S. pyogenes* (0.33 ± 1.15 mm) in the disk diffusion method. Das et al. [36] measured inhibition zones for *B. cereus* (35 mm), *C. albicans* (16 mm), and *R. oligosporus* (15 mm). Both works detected very high zones of inhibition compared to our work, but the antimicrobial activity of PCEO depends on various factors (composition, origin, preparation) of the essential oil as well as on differences in the methodological procedure.

Paulus et al. [26] analysed the antimicrobial effect of the PCEO essential oil and found that very low concentrations were needed to inhibit microorganisms, determining a minimum inhibitory concentration of about 0.195 μL/mL for most tested microorganisms. Chakrapani et al. [32] determined MICs for *B. subtilis*, *E. coli*, *S. aureus,* and *E. aerogenes* in the range of 40 to 80 μL. The Wong [37] agar microdilution test of *E. coli* and *S. epidermidis* showed that *Pogostemon cablin* extract was bacteriostatic in *E. coli* with a MIC of 1.66 mg/mL and in *S. epidermidis* with a MIC greater than 1.66 mg/mL. Adhavan et al. [38] observed the antimicrobial effect of PCEO against *C. albicans* at a concentration of 25 mg/mL. Das et al. [36] determined MICs for *B. cereus* (250 μg/mL), *C. albicans* (750 μg/mL), and *R. oligosporus* (250 μg/mL). Despite the different expressions of the result, the authors agree with our findings that relatively low minimum inhibitory concentrations are needed.

The increasing antibiotic resistance of biofilm has drawn attention to the study of alternative sources of antibiofilm agents [39,40]. Many authors analyse essential oils as potential antibiofilm agents [16,21,41–43]. To date, a very limited amount of work has been done on the antibiofilm activity of PCEO. Nithyanand et al. [44] discovered the potential of PCEO as an antibiofilm agent. Junren et al. [45] detected the antibiofilm effect of aqueous extracts from *P. cablin*. Snoussi et al. [46] states that essential oils are suitable alternatives for biofilm inhibition. Pereira et al. [47] confirmed the effectiveness of MALDI-TOF MS compared to electron microscopy for biofilm structure analysis. Li et al. [48] used MALDI-TOF MS to analyze *B. subtilis* biofilm and determine the spatial distribution of specific peptides and lipopeptides. Stîngu et al. [49] state that financially, materially, time-consuming, and technically demanding methods with low reproducibility are often needed for the analysis of biofilms. The biofilm analysis method using MALDI-TOF MS Biotyper provides an innovative, timesaving, cost-effective, and accurate tool for studying biofilms.

The effect of PCEO in the steam phase on a food model has not yet been studied. Kloucek et al. [50] reported that the use of the vapor phase of essential oils reduces the concentrations required to inhibit microorganisms compared to the contact application of the liquid phase. Garzoli et al. [51] compared the effect of contact antimicrobial activity and vapor-phase antimicrobial activity and found that vapor phases were more effective than liquid phases. Znini [52] states that according to the results obtained in their study, the test of essential oils in the vapor phase shows better antifungal activity against the tested pathogens than those observed in the liquid phase. Some other studies also confirm that essential oils at the vapor phase are more effective than contact application [53–56]. Nadjib et al. [57] state that this may be due to the lipophilic molecules in the liquid phase combining to form micelles, thus preventing the essential oil from adhering to microorganisms, with the vapor phase allowing free adhesion to microorganisms, which increases its effectiveness.

Many authors have observed contact toxicity of PCEO, but very few authors have tested vapor phase toxicity. Rocha et al. [58] recorded a lethal and sublethal effect of PCEO on leaf-cutting ants. Chen et al. [59] reported that PCEO shows significant repellent, contact toxicity, and anti-sedative activity against *M. persicae*. The toxicity of PCEO has been observed against *Spodoptera exigua*, termites, and flies [60]. Albuquerque et al. [61] observed a repellent effect of PCEO against two species of ants. Pavela [62] experienced a strong insecticidal effect of PCEO at higher concentrations against *Musca domestica*. Zeng et al. [63] observed that PCEO showed strong contact toxicity against *Pieris rapae* L. Liu et al. [27] found a strong repellent effect of PCEO against *Blattella germanica*.

## 5. Conclusions

PCEO has a diverse chemical composition, and its antioxidant activity was high. PCEO had relatively long-term antibacterial effects as well as antibiofilm effects, as observed on various surfaces and determined by MALDI-TOF MS Biotyper. The antifungal activity of *P. cablin* was very weak during contact application, but significantly higher antifungal activity was observed in the in situ vapor phase test on the food model. No significant increases in efficacy were observed in the in situ antibacterial test in the vapor phase. In the future, PCEO could be used as an antifungal agent in the storage of bakery products and other food industries items. It could also be used for storing root vegetables to protect crops from mold. In the future, it is necessary to verify the effect of the essential oil on the sensory properties of food and crops. The insecticidal activity was observed only at the higher concentrations of PCEO tested. The insecticidal activity of PCEO on other insect species needs to be tested and could subsequently be used as an insecticidal agent.

**Author Contributions:** Conceptualization, L.G., P.B. and M.K.; methodology, L.G., P.B., J.Š., V.V., H.Ď., N.L.V., M.V. and M.K.; software, L.G., V.V. and M.K.; validation, L.G., P.B. and M.K.; formal analysis, L.G., P.B. and M.K.; investigation, L.G., P.B., V.V., J.Š., H.Ď., N.L.V., M.V. and M.K.; resources, M.K.; data curation, L.G. and M.K.; writing-original draft preparation, L.G., P.B. and M.K.; writing-review and editing, L.G., P.B. and M.K.; visualization, L.G.; supervision, M.K.; project administration,

M.K.; funding acquisition, M.K. All authors have read and agreed to the published version of the manuscript.

**Funding:** This research was funded by the grant APVV-20-0058 "The potential of the essential oils from aromatic plants for medical use and food preservation".

**Institutional Review Board Statement:** Not applicable.

**Informed Consent Statement:** Not applicable.

**Data Availability Statement:** Not applicable.

**Acknowledgments:** This work was supported by the grants of the VEGA no. 1/0180/20.

**Conflicts of Interest:** The authors declare no conflict of interest.

## Appendix A

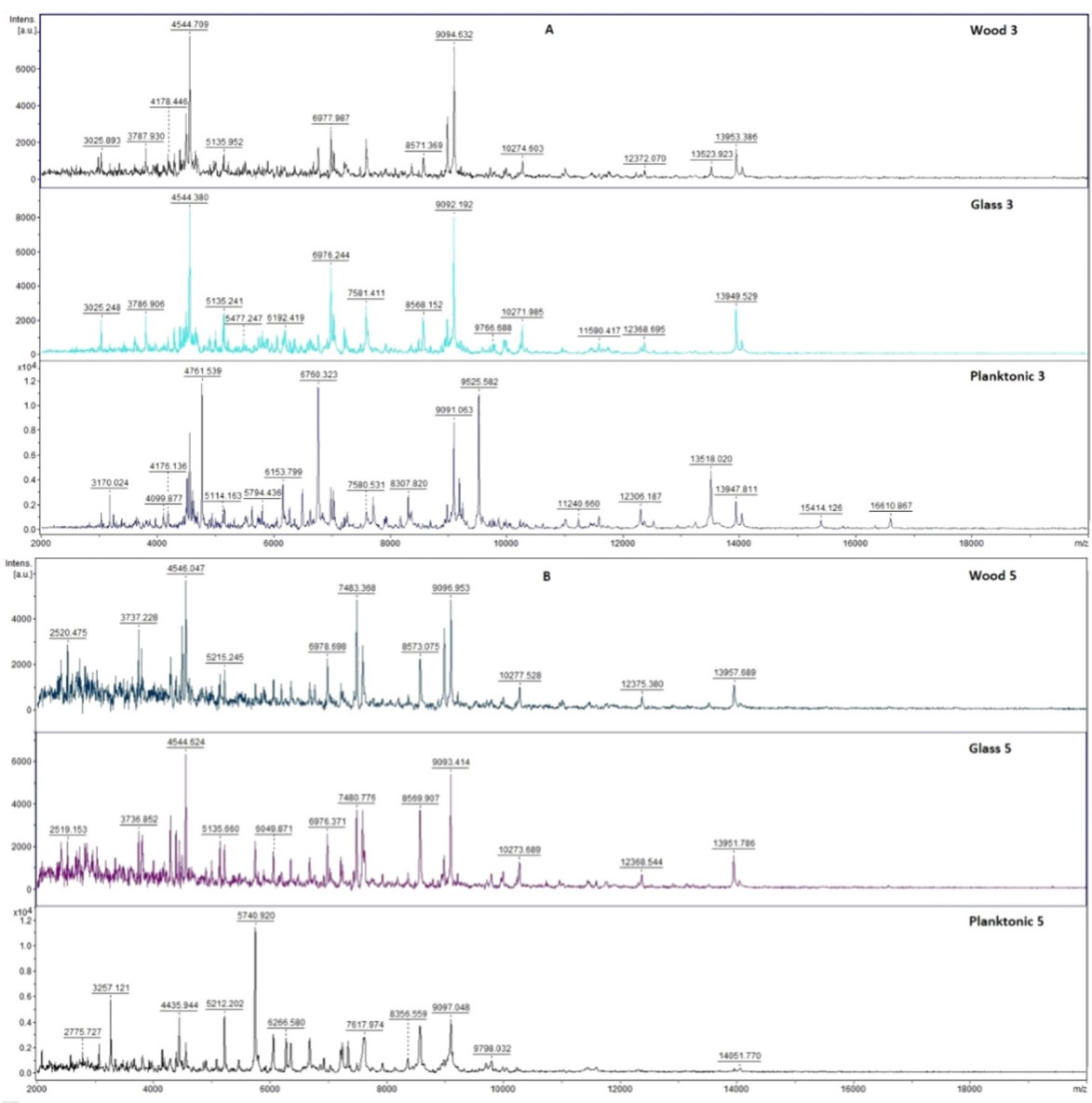

**Figure A1.** *Cont*.

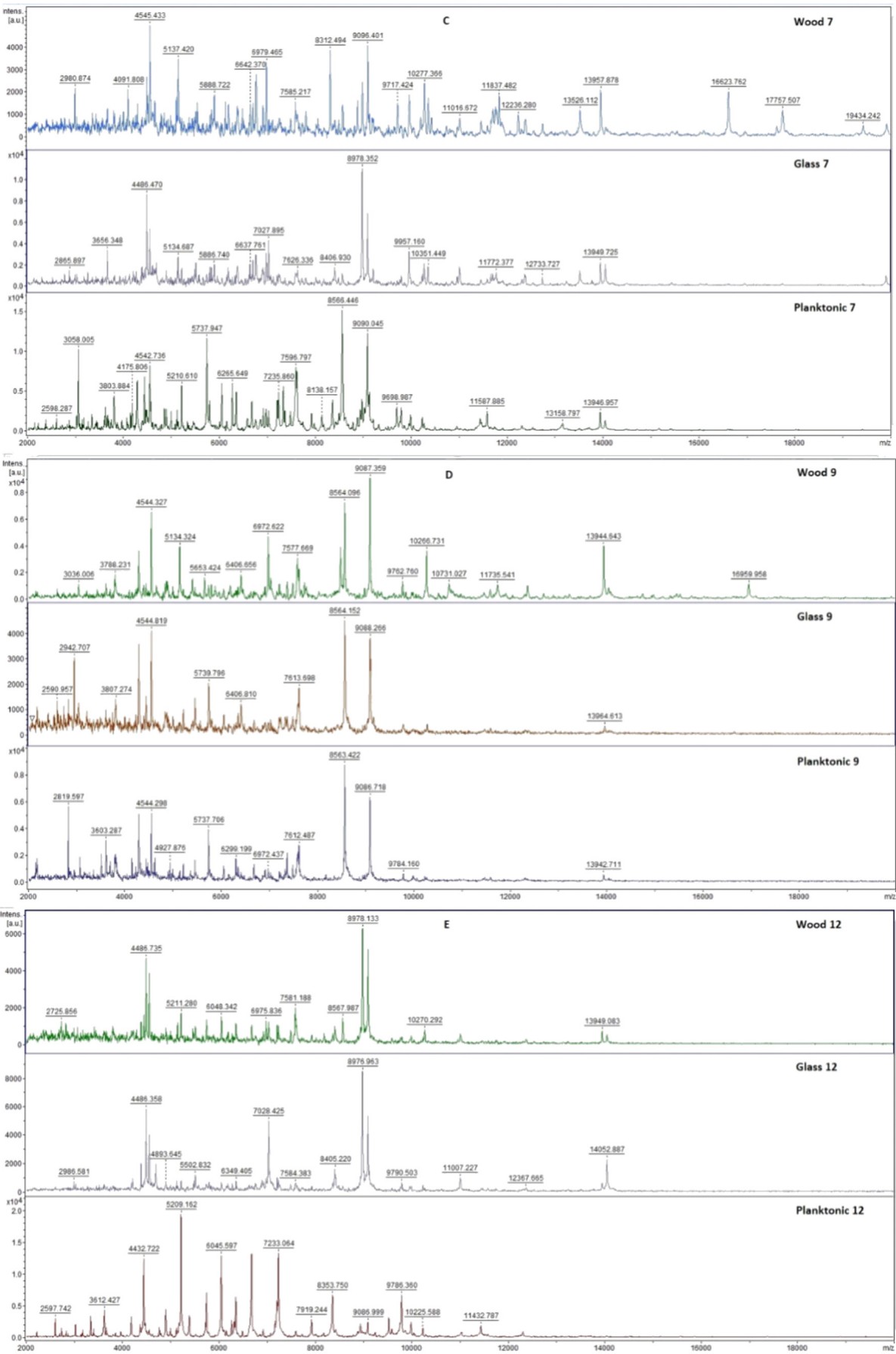

**Figure A1.** *Cont.*

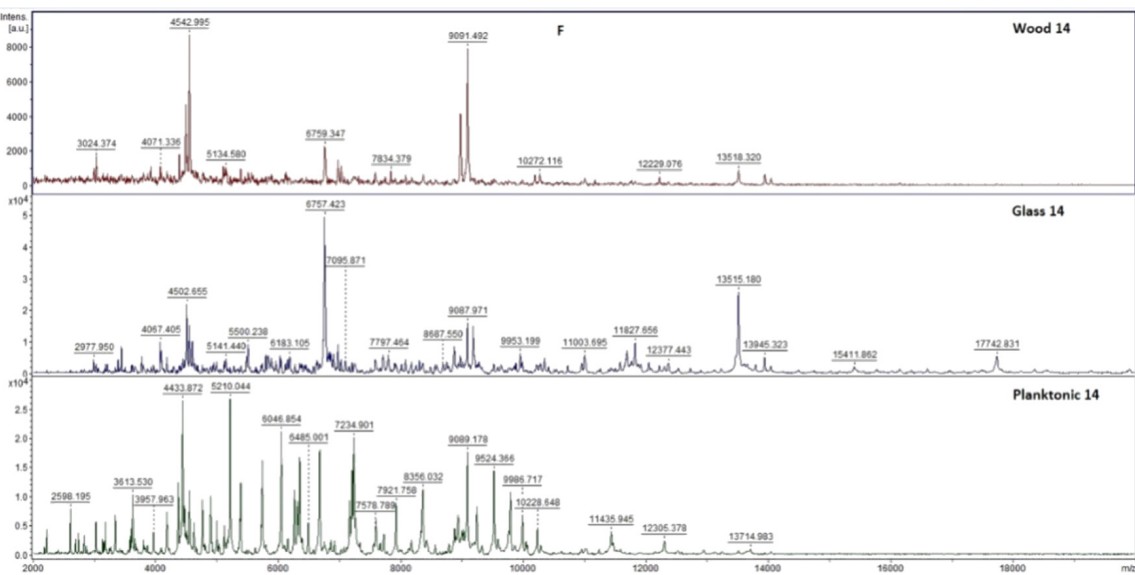

**Figure A1.** MALDI-TOF mass spectra of *P. fluorescens* biofilm during development after the addition of PCEO: (**A**) 3rd day, (**B**) 5th day, (**C**) 7th day, (**D**) 9th day, (**E**) 12th day, and (**F**) 14th day.

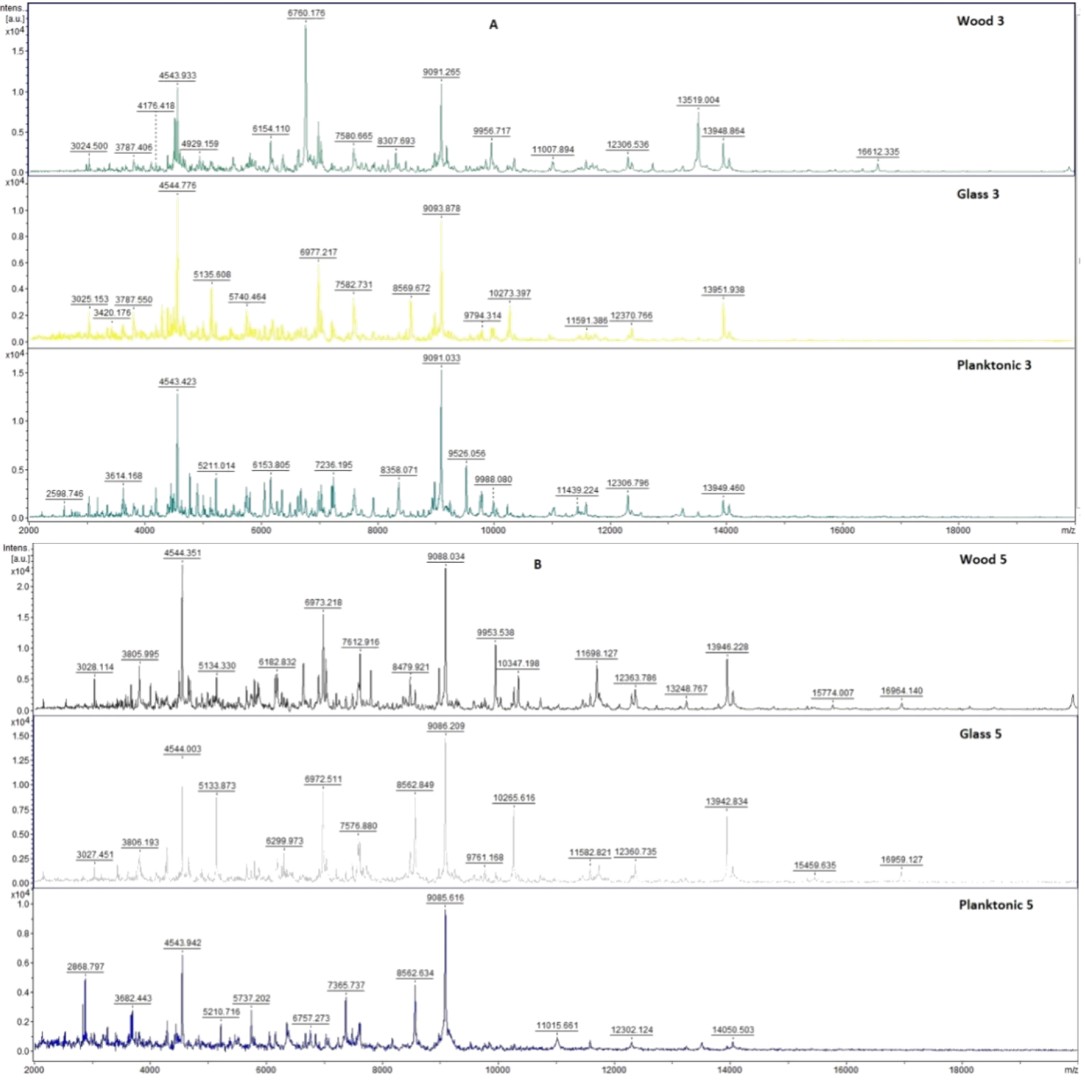

**Figure A2.** *Cont*.

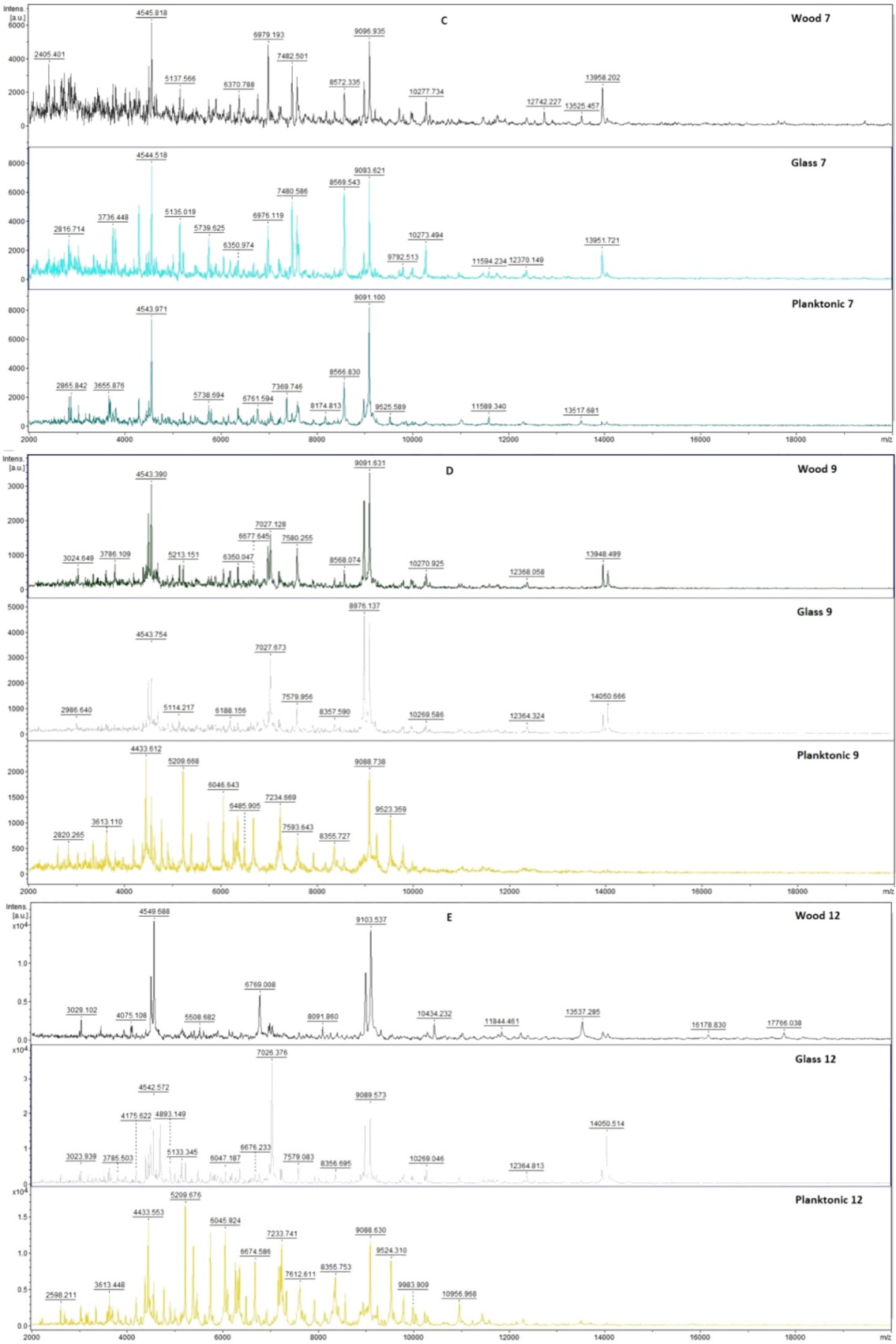

**Figure A2.** *Cont.*

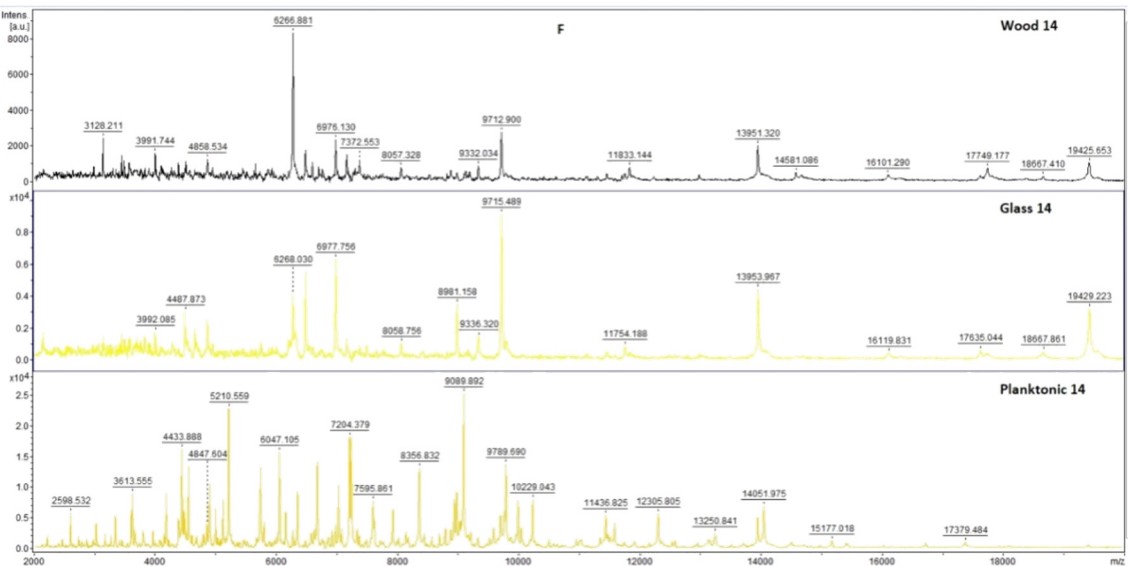

**Figure A2.** MALDI-TOF mass spectra of *S. enteritidis* biofilm during development after the addition of PCEO: (**A**) 3rd day, (**B**) 5th day, (**C**) 7th day, (**D**) 9th day, (**E**) 12th day, and (**F**) 14th day.

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
