# Peer review of "Biological Activity of Pogostemon cablin Essential Oil and Its Potential Use for Food Preservation"

_agronomy, doi:10.3390/agronomy12020387_

Round 1

Reviewer 1 Report

The aim of this research was to study the biological activity of Pogostemon cablin essential oil and its potential use for food preservation. It is an interesting topic and authors managed to provide a clear statement of the scientific area, a range of research on the topic, critically analyse the selected topic using published data and provide an indication of what further research is necessary.

Author Response

Reviewer #1

The aim of this research was to study the biological activity of Pogostemon cablin essential oil and its potential use for food preservation. It is an interesting topic and authors managed to provide a clear statement of the scientific area, a range of research on the topic, critically analyse the selected topic using published data and provide an indication of what further research is necessary.

Response: Thank you very much for the positive evaluation and for your time.

Reviewer 2 Report

The manuscript entitled "Biological activity of Pogostemon cablin essential oil and its potential use for food preservation" is generally a good one. But in my opinion it needs to be improved in order to be published.
Below are my suggestions for improvement:
The purpose of this paper is missing from the abstract
Materials and methods: This section is well written and well explained, in order that other researchers can replicate the methods if the case arrives.
Table 1 overlaps the line numbers
Table 1 is not a statistical analysis
There is no statistical analysis in Tables 2,3, 4 either. The standard deviation is not enough
Tables 5-10 have the corresponding statistical analysis
Discussion: As for the discussion, it is a good discussion, but not enough information to correlate the results with existing references. I suggest you improve this section.
There are no studies on the activity of preservatives in the food industry, it is just a potential conclusion. Consequently, the title may mislead future readers.

Author Response

Reviewer #2

The manuscript entitled "Biological activity of Pogostemon cablin essential oil and its potential use for food preservation" is generally a good one. But in my opinion, it needs to be improved in order to be published. Below are my suggestions for improvement:

Point 1: The purpose of this paper is missing from the abstract.

Response: Thank you very much for this favorable opinion. We tried improving our paper according to your remarks. The aims of our work were added to the abstract.

Point 2: Materials and methods: This section is well written and well explained, in order that other researchers can replicate the methods if the case arrives.

Response: Thank you very much for this favorable opinion.

Point 3: Table 1 overlaps the line numbers.

Response: Thank you very much for this favorable opinion. We tried improving our paper according to your remarks. The table has been modified so that it does not overlap row numbers.

Point 4: Table 1 is not a statistical analysis. There is no statistical analysis in Tables 2,3, 4 either. The standard deviation is not enough. Tables 5-10 have the corresponding statistical analysis.

Response: Thank you very much for this opinion, but it is not possible to perform a statistical analysis other than the standard deviation within the analyzed data shown in Tables 1 to 3. These are different types of microorganisms and one type of essential oil, and statistical significance cannot be assessed, because different results of antimicrobial activity of different microorganisms can be evaluated together in this example of table. Statistical significance can only be assessed for different essential oils on the same microorganisms or at different essential oil concentrations. Statistical significance was added to Table 4.

Point 5: Discussion: As for the discussion, it is a good discussion, but not enough information to correlate the results with existing references. I suggest you improve this section.

Response: Thank you very much for this favorable opinion. We tried improving our paper according to your remarks. The discussion was developed and supplemented.

Point 6: There are no studies on the activity of preservatives in the food industry, it is just a potential conclusion. Consequently, the title may mislead future readers.

Response:  Thank you very much for this favorable opinion. The title of our work states that it is a potential use for food preservation. This emphasizes that it is not a preservative but, due to its properties, could be used in the future for the food preservation.

Reviewer 3 Report

The present manuscript is dedicated to the essential oil of Pogostemon cablin (patchouli) and its potential in food preservation. The results are of interest but the article should be focused on the new findings. The essential oil of patchouli, its chemical composition and antimicrobial properties are well known, there are numerous publications on the subject. For this reason, the Authors should very clearly outline what is new in their work and discuss it in depth: the antibiofilm activity and the effect of the essential oil in the steam phase on food models. There are further points which need further attention:

  1. The first and the second sentences of the Abstract are contradictory, please change.
  2. 1 and 3. should be presented as Complementary material.
  3. Listing the literature data about the chemical composition of Pogostemon cablin essential oil cannot be regarded as a discussion on the topic. The same is true for the literature data on the antimicrobial activity.
  4. The sentences in rows 507 – 509 are results, they do not belong to conclusions.

Author Response

Reviewer #3

The present manuscript is dedicated to the essential oil of Pogostemon cablin (patchouli) and its potential in food preservation. The results are of interest but the article should be focused on the new findings. The essential oil of patchouli, its chemical composition and antimicrobial properties are well known, there are numerous publications on the subject. For this reason, the Authors should very clearly outline what is new in their work and discuss it in depth: the antibiofilm activity and the effect of the essential oil in the steam phase on food models. There are further points which need further attention:

Point 1: The first and the second sentences of the Abstract are contradictory, please change.

Response: Thank you very much for this favorable opinion. The sentences in question have been changed.

Point 2: 1 and 3. should be presented as Complementary material.

Response: Thank you very much for this favorable opinion. We tried improving our paper according to your remarks. Figures 1 and 3 are included as supplementary material.

Point 3:

Listing the literature data about the chemical composition of Pogostemon cablin essential oil cannot be regarded as a discussion on the topic. The same is true for the literature data on the antimicrobial activity.

Response: Thank you very much for this favorable opinion. We tried improving our paper according to your remarks. The discussion was developed and supplemented.

Point 4: The sentences in rows 507 – 509 are results, they do not belong to conclusions.

Response: Thank you very much for this favorable opinion. We tried improving our paper according to your remarks.

Round 2

Reviewer 2 Report

The authors responded to all my comments.

Reviewer 3 Report

The Authors have answered the questions and made the suggested amendments.